# Inorganic Boron-Based Nanostructures: Synthesis, Optoelectronic Properties, and Prospective Applications

**DOI:** 10.3390/nano9040538

**Published:** 2019-04-03

**Authors:** Yan Tian, Zekun Guo, Tong Zhang, Haojian Lin, Zijuan Li, Jun Chen, Shaozhi Deng, Fei Liu

**Affiliations:** State Key Laboratory of Optoelectronic Materials and Technologies, Guangdong Province Key Laboratory of Display Material and Technology, School of Electronics and Information Technology, Sun Yat-sen University, Guangzhou 510275, China; tiany29@mail2.sysu.edu.cn (Y.T.); guozk@mail2.sysu.edu.cn (Z.G.); zt423109972@163.com (T.Z.); linhjian@foxmail.com (H.L.); lizj35@mail2.sysu.edu.cn (Z.L.); stscjun@mail.sysu.edu.cn (J.C.); stsdsz@mail.sysu.edu.cn (S.D.)

**Keywords:** inorganic boron-based nanostructures, boron monoelement nanowire and nanotube, borophene, rare-earth boride (REB_6_), optoelectronic properties

## Abstract

Inorganic boron-based nanostructures have great potential for field emission (FE), flexible displays, superconductors, and energy storage because of their high melting point, low density, extreme hardness, and good chemical stability. Until now, most researchers have been focused on one-dimensional (1D) boron-based nanostructures (rare-earth boride (REB_6_) nanowires, boron nanowires, and nanotubes). Currently, two-dimensional (2D) borophene attracts most of the attention, due to its unique physical and chemical properties, which make it quite different from its corresponding bulk counterpart. Here, we offer a comprehensive review on the synthesis methods and optoelectronics properties of inorganic boron-based nanostructures, which are mainly concentrated on 1D rare-earth boride nanowires, boron monoelement nanowires, and nanotubes, as well as 2D borophene and borophane. This review paper is organized as follows. In Section I, the synthesis methods of inorganic boron-based nanostructures are systematically introduced. In Section II, we classify their optical and electrical transport properties (field emission, optical absorption, and photoconductive properties). In the last section, we evaluate the optoelectronic behaviors of the known inorganic boron-based nanostructures and propose their future applications.

## 1. Introduction

Boron is very special, because it is the only nonmetallic element in group III, the lightest nonmetallic element in the periodic table, and excellent properties similar to carbon. Moreover, boron possesses unique physical and chemical properties due to the B_12_ icosahedra structural unit consisting of a particular three-center two-electron bond, such as high melting point, low density, extreme hardness, and nice chemical stability [1,2,3,4]. Over the past few decades, boron has attracted much attention from researchers, and boron-based inorganic compounds are popular lightweight structural materials and thermionic cathode materials [5,6]. As one of the greatest scientists of the 20th century, Dr. Lipscomb was respectively awarded the 1976 and 1979 Nobel Prizes in Chemistry for his outstanding contribution to the configuration study of borane and excellent research on metal borides [7]. The occurrence of these two important achievements heralded the first hot wave research of boron. However, over the next 20 years, the research of the boron-based materials gradually tended to stabilize. As the building blocks for the nanodevices, nanomaterials (including two-dimensional (2D) layered structures, one-dimensional (1D) nanowires and nanotubes, and zero-dimensional (0D) nanoparticles) have attracted more and more attention [8,9]. Compared with their corresponding bulk counterparts, inorganic boron-based nanomaterials exhibit more fascinating optical and electrical transport behaviors due to their higher aspect ratio, larger specific surface area, and smaller size. Among them, rare-earth boride (REB_6_) nanomaterials (LaB_6_, CeB_6_ etc.) are excellent cold cathode candidates, since they have low work function, ultrahigh hardness, and a high melting point, as well as excellent electrical and thermal conductivity performances [10,11,12,13,14,15,16]. Being typical Kondo topological insulators, SmB_6_, CeB_6_, and YbB_6_ nanomaterials have attracted the researchers since the 2016 Nobel Prize in Physics was awarded to David J. Thouless, F. Duncan M. Haldane, and J. Michael Kosterlitz for their theoretical discoveries on the topological phase transitions and topological phases of matter [17,18,19,20,21,22]. At the end of 2015, the 2D layered structure of the boron single element was first synthesized experimentally, and was named borophene [23,24,25]. Borophene has been theoretically proposed to have numerous excellent and unique properties, which may have a promising future in batteries, flexible devices and high-speed electronic devices [26,27,28]. As a result, the relevant studies on inorganic boron-based nanomaterials resonate in the material sciences and condense physics fields again.

In this paper, we review the current developments of inorganic boron-based nanostructures (boron monoelement and rare-earth borides). First, we introduce their synthesis methods. Second, the optoelectronic properties of the boron-based nanostructures are classified. Finally, we give the evaluations on the optoelectronic performances of inorganic boron-based nanostructures and propose their applications based on our research results.

## 2. Synthesis Methods of Inorganic Boron-Based Nanostructures

### 2.1. One-Dimensional Boron-Based Nanomaterials

One-dimensional (1D) nanostructures offer ideal platforms to study the dependence of electrical transport, thermal conductivity, and mechanical strength on the dimensionality or size effects [29]. In general, 1D nanostructures are widely used in field-effect transistors, light-emitting diodes (LEDs), photodetectors, solar cells, field emission displays, and sensors [29,30,31,32,33,34,35,36]. In 2010, we introduced 1D boron single element nanostructures [37]. However, to date, although the synthesis methods of 1D inorganic boron-based nanostructures (nanowires, nanobelts, and nanotubes) have been well described, they need to be further renewed to meet the requirements of their rapid developments.

#### 2.1.1. Rare-Earth Boride Nanostructures

There are usually six kinds of stoichiometry for rare earth borides, namely REB_2_, REB_4_, RE_2_B_5_, REB_6_, REB_12_, and REB_66_, in which RE represents rare-earth element (La, Sm, Ce, Eu, Gd, etc.). Among them, REB_6_ is very popular, and has been widely used in commercial cathode materials. The typical lattice structure of REB_6_ is given in Figure 1. REB_6_ belongs to the cubic lattice, in which boron octahedrons consisting of six boron atoms are located at the vertex positions of the cubic lattice, and rare-earth atoms occupying the central position of the cubic lattice form the other body-centered CsCl-type cubic lattice. 

In actual field emission (FE) applications, higher crystallinity, lower work function, and larger aspect ratio are highly demanded for REB_6_ nanomaterials, which are believed to be the crucial criterions to evaluate the quality of cold cathode nanomaterials. The synthesis methods of REB_6_ nanostructures can be classified into three types (Table 1): chemical vapor deposition (CVD), hydrothermal reaction, and electrochemical etching methods. Chi and Fan et al. [38,39,40,41,42,43] respectively applied a no-catalyst CVD method to synthesize different REB_6_ (PrB_6_, NdB_6_, CeB_6_, GdB_6_, and EuB_6_) nanowires at 1000–1150 ℃ by choosing rare-earth metal powders and B-contained gas (BCl_3_ or B_2_H_6_) as source materials. As seen in Figure 2a,c, single crystalline PrB_6_ and NdB_6_ nanowires have diameters of several tens of nanometers and lengths extending to several micrometers as well as uniform diameter along the growth axis. Using the CVD method, Xu et al. [35] reported the successful growth of 1D LaB_6_ nanoneedles with a diameter decreasing from the bottom to the top and nanowires with uniform diameter by changing the distance between the precursors and substrate. By the hydrothermal reaction method, Han et al. [39] obtained single crystalline SmB_6_ nanowires growing along the [001] direction, whose averaged diameter and length were respectively 80 nm and 4 μm. To prepare REB_6_ nanomaterials, nearly all of the researchers used toxic, inflammable, or expensive gas as the source materials, which inevitably bring some threat to our lives. To solve the above problem, our group developed a simple solid-source CVD method to synthesize single crystalline SmB_6_ and LaB_6_ nanowire arrays using Ni catalysts. In our method, the low-cost, nontoxic solid B, B_2_O_3_, and LaCl_3_ powders or Sm film were the source materials, which can avoid the potential pollution of the nanomaterial production [44,45]. In comparison with other two ways, the CVD method is more convenient for the growth of high-quality REB_6_ nanomaterials, because the control of their morphology, crystal structure, and stoichiometry are relatively easier.

#### 2.1.2. Boron Single Element Nanostructures

Boron nanostructures have potential applications in ideal cold-cathode electron sources, high-temperature semiconductor devices, and field-effect transistors, which have attracted much interest from researchers.

##### Boron Nanowires

Magnetron sputtering, laser ablation, thermal carbon reduction, thermal evaporation, and chemical vapor methods have been utilized to prepare amorphous or crystalline boron nanowires (BNWs) [46,47,48,49,50]. The morphology and crystallinity of the as-synthesized BNWs can differ with the growth methods. By the combination of CVD and ultraviolet photoresist technology, large-scale patterned boron nanowire (BNW) arrays were synthesized on Si substrate [51]. As found in Figure 3a,b, all of the nanowire patterns exhibit a regular square with a size of 25 μm × 60 μm, and the BNWs in each patterns have a uniform morphology, which may be more beneficial for future integrated circuit device applications. Using magnetron sputtering, Cao et al. [46] successfully prepared amorphous boron nanowire aligned arrays on silicon substrates. The as-grown BNWs have diameters from 20 to 80 nm and lengths up to several tens of micrometers, which are indexed as amorphous structures (Figure 3c,d).

##### Boron Nanotube

Boron nanotubes (BNTs) have been theoretically proposed to have metallic properties, whether their structure configurations are armchair or zigzag. In 2004, the magnesium-substituted mesoporous silica template (Mg–MCM-41) method was applied to prepare pure boron single-wall nanotubes at 870 °C according to the method by Ciuparu et al. [52], using the mixture of BCl_3_ and H_2_ as gas sources. The authors attributed the Raman peaks at about 210 cm^−1^ (peak a) and 430 cm^−1^ (peak b) to typical tubular structures (Figure 4a), corresponding to the characteristic radial breathing mode. Using boron (99.99%) and boron oxide powders (99.99%) as source materials, our group reported the first large-scale fabrication of single crystalline multilayered BNTs in 2010 [53]. As seen in Figure 4b–e, the as-synthesized BNTs have lengths of several micrometers and diameters of about 30 nm. Moreover, these nanotubes are indexed as multilayered single crystalline boron nanotubes with the interlayer spacing of about 3.2 Å. Most of all, these nanotubes were experimentally proven to have metallic properties as theoretical predictions, which didn’t vary with their chirality.

At the viewpoint of future cold cathode applications of 1D boron nanostructures, the CVD method has advantages over other synthesis methods owing to its better morphology, aspect ratio, and crystallinity controllability. In addition, large-scale, high-quality 1D boron-based nanostructure arrays are much easier to obtain on rigid or flexible substrates by the CVD method, which is very beneficial for their cell or photosensitive device applications.

### 2.2. Two-Dimensional Boron-Based Nanomaterials

Inspired by the stunning achievements of 2D materials [54], much effort has been devoted to exploring 2D boron-based nanostructures in both theoretical calculations and experiments [55,56,57]. To date, there are three synthesis methods of 2D boron-based nanostructures: molecular beam epitaxy (MBE), liquid-phase exfoliation, and CVD methods. The researchers are devoted to modulating the morphology, layer number, and structure configuration of 2D boron-based nanostructures due to the effects on their physical properties.

#### 2.2.1. Two-Dimensional Boron Monoelement Nanostructures

By MBE, Mannix and Wu [24,25] respectively reported the first growth of atomically thin, crystalline 2D boron sheets on silver surfaces under an ultrahigh-vacuum system. As shown in Figure 5a–c, Mannix et al. [24] obtained the atomic image of borophene, and ascertained that it has the same metallic and highly anisotropic characteristics as theoretically predicted. Wu et al. [25] uncovered two allotropes of boron sheets on Ag (111) substrate, which are respectively β_12_ and χ_3_ sheets. Both have a similar triangular lattice but different arrangements of periodic holes, as revealed by STS spectra (Figure 5d–f). Bedzyk’s research further proved that atomically thin borophene sheets on the Ag (111) surface have a Van der Waals-like structure [58]. By X-ray standing wave-excited X-ray photoelectron spectroscopy, the lattice positions of boron atoms with multiple chemical states were confirmed, revealing that the thickness of a single layer borophene is 2.4 Å on an unreconstructed Ag surface. Afterwards, Wu et al. [59] synthesized a pure honeycomb, graphene-like borophene on the Al (111) substrate by MBE. Moreover, their theoretical calculations discovered that the honeycomb borophene is stable on the Al (111) surface, because there exists one electron transfer from the Al (111) substrate to each boron atom, which stabilizes the structure. The atomic resolution STM images of the honeycomb borophene are shown in Figure 6, in which the lattice constant is about 0.29 nm. Very recently, single crystalline borophene with an area up to 100 μm^2^ has been fabricated on the Cu (111) substrate via the MBE method by Gozar et al. [60]. The as-grown borophene was composed of novel triangular networks with a concentration of hexagonal vacancies of η = 1/5, belonging to β_12_ and χ_3_ phases.

By CVD, Tai et al. also prepared 2D γ-boron films on copper foil using boron and boron oxide powders as the source materials [23], as seen in Table 2. The as-prepared γ-B_28_ film was identified as a semiconductor with a direct bandgap of around 2.25 eV, which had a strong photoluminescence emission band at 626 nm (Figure 7). Besides, Tsai et al. [61] applied a B plasma-assisted technique to deposit a multilayer β-borophene on an insulating SiN_x_ film on Si substrate.

Using the sonication-assisted liquid-phase exfoliation method, Teo et al. [62] obtained high-yield few-layer B sheets. As given in Figure 8, the as-synthesized few-layer B sheets have a clear diffraction pattern with a d spacing of 0.504 nm, conforming to the (104) plane of β-rhombohedral B crystal.

#### 2.2.2. Two-Dimensional Boron-Based Nanostructures

Due to the structural instability of borophene, the fully-hydrogenated borophene (i.e., borophane) with enough stability has aroused much interest, which was theoretically predicted by Xu et al. [63]. By the first-principle calculations, borophane is anticipated to have a stable surface configuration due to electron transfer after full hydrogenation, as presented in Figure 9. Moreover, the Fermi velocity of borophane is as high as 3.5 × 10^6^ m s^−1^ under HSE06 level, which is the highest among the known 2D materials, and is four times larger than that of graphene. Although there are many theoretical calculations and predictions on borophane [64,65], it has not been experimentally synthesized. On the other hand, the 2D boron oxide (BO) structure is also proposed to be stable after experiencing a suitable stoichiometric degree of oxidation. Yang et al. [66] studied the structural configurations of the 2D boron oxide sheet by using the first-principle global optimization method, suggesting that they have potential applications in high-speed nanoelectronic devices. Also, Zhang et al. [67] investigated the physical properties of the hexagonal borophene oxide sheets in Wu’s work [59], revealing they should possess remarkable mechanical and electronic properties. Furthermore, 2D borophene oxide nanostructures with completely flat configuration may serve as a promising platform for studying 2D topological phases, because its energy band hosts a nodal loop centered around the Y point in the Brillouin zone, and exhibits different topological indices before and after transition [67].

As mentioned above, the MBE way is more suitable for the fabrication of a monolayer or few-layer high-crystallinity borophene, while CVD method has advantages on the layer or structure configuration control of large-area borophene with different standing directions to the substrate. For high-yield borophene, we suggest adopting the sonication-assisted liquid-phase exfoliation method, which has a more promising future in industry. Until now, only a few phases of borophene have been successfully prepared, which is far less than the several dozen phases in theoretical prediction. As a result, it still remains a big challenge for the researchers.

## 3. Optoelectronic Properties

Inorganic boron-based nanostructures have potential applications in optoelectronic nanodevices because of their particular physical and chemical properties. In this section, we review the electrical and optical properties measured on inorganic boron-based nanostructures in the past two decades.

### 3.1. Electrical Properties

#### 3.1.1. Field Emission (FE) Properties

Inorganic boron-based nanostructures have been proposed as ideal cathode choices because they have small surface electron affinity, high electrical and thermal conductivity, a large aspect ratio, a high melting point, and large current endurance. Xu et al. [36] reported that the 1D LaB_6_ nanostructures produced by CVD exhibited good FE characteristics, which had a turn-on field of 1.82 V μm^−1^ and a threshold field of 2.48 V μm^−1^. In our recent works, the LaB_6_ nanowire arrays had a low turn-on (2.2 V μm^−1^) and threshold field (2.9 V μm^−1^) as well as nice field emission (FE) stability with a current fluctuation of only 1.7% [44]. Further research showed that individual LaB_6_ nanowires can bear the maximum current of 96 μA and the maximum emission current of 0.3 cm^2^ LaB_6_ nanowire film can reach as high as 5.6 mA. Most of all, the LaB_6_ nanowire film exhibited the recoverable emission performance after O_2_ absorption or desorption, making them suitable for applications in air. Recently, Zhang et al. [68] measured the working performance of cold field emission SEM equipped with a single LaB_6_ nanowire emitter (Figure 10). The typical morphology of the LaB_6_ nanowires (NWs) was presented in Figure 10a. Figure 10b schematically illustrates the fabrication of the LaB_6_ emitter. An etched W tip is manipulated to approach individual LaB_6_ nanowires, and subsequently, it is handled by an ion beam to form a tight contact with the LaB_6_ nanowire via Van der Waals force. After the LaB_6_ nanowire breaks off from the substrate and adheres onto the tip, the nanowire is fixed on a carbon pad by the electron beam-induced deposition (EBID) technique and the fabrication of the LaB_6_ emitter is accomplished. In these FE experiments in the modified SEM system [68], the electron gun equipped with the LaB_6_ NW displays an ultrahigh angular current density of ~2.4 × 10^5^ μA sr^−1^, which is 1000 times larger than the electron gun with the W (310) emitter (Figure 10c). Additionally, the current fluctuation (0.32%) for the LaB_6_ emitter is obviously lower than that for the W (310) emitter (7.2%) under the same experimental conditions. The signal noise comparison for imaging the acceleration electrode using the LaB_6_ NW and W (310) emitters are given in Figure 10d, in which the resolution and image quality of the LaB_6_ emitter are obviously higher than those of the W emitter. Considering that the acquisition image time of the W emitter is twice as long as that of the LaB_6_ nanowire emitter, the LaB_6_ nanowire emitter should have more advantages than the commonly-used W (310) emitter. Also, the FE properties of the SmB_6_ nanowires were compared with those of the SmB_6_ nanopencils, which have a thick bottom and a sharp tip, to evaluate their application in cold cathodes [45]. The SmB_6_ nanowires had a lower turn-on field of 6.5 V μm^−1^ in comparison with the SmB_6_ nanopencils (6.9 V μm^−1^), and their maximum emission current density can reach several hundred µA cm^−2^ (Figure 11).

Similarly, the BNWs exhibited good emission properties [51]. The turn-on and threshold fields of large-area BNW patterns are respectively 4.3 V μm^−1^ and 10.4 V μm^−1^, as provided in Figure 11e,f. In addition, they exhibited a very uniform emission image (inset), in which the distribution uniformity of the emission patterns was 81.8%, and their brightness distribution uniformity was over 88.9%. The FE behaviors of boron-based nanostructures can compare favorably to those of many nanomaterials with excellent emission performance, which suggests that they have potential applications in cold cathode electron sources (Table 3).

#### 3.1.2. Capacitance Characteristics

Inorganic boron-based nanostructures also have great potential in supercapacitors. Zhi and Liu et al. [73] stated that elemental boron-based supercapacitors exhibited excellent performances whether they were placed in all alkaline, neutral, or acidic electrolytes. Boron nanowire–carbon fiber cloth (BNWs–CFC) electrodes achieve a capacitance up to 42.8 mF cm^−2^ at a scan rate of 5 mV s^−1^ and 60.2 mF cm^−2^ at a current density of 0.2 mA cm^−2^ in acidic electrolyte (Figure 12), respectively. In addition, the BNWs–CFC electrodes still retain high performance even after being bent 1000 times, revealing that they possess excellent mechanical properties. Subsequently, the LaB_6_ nanowires electrode had an areal capacitance as high as 17.34 mF cm^−2^ in 1.0 M of Na_2_SO_4_ solution at a current density of 0.1 mA cm^−2^ and 16.03 mF cm^−2^ at a scan rate of 5 mV s^−1^. Moreover, the LaB_6_ nanowires electrodes displayed outstanding cycling stability after 10,000 charging/discharging cycles, which suggests that they are highly effective electrode materials for supercapacitors [74]. Soon afterwards, Teo et al. [62] used the exfoliated few-layer B sheets as the electrode materials of a supercapacitor. The supercapacitor using B-sheet electrodes exhibited impressive electrochemical performance with a wide potential window up to 3.0 V. Moreover, the energy density of the supercapacitor can reach as high as 46.1 W kg^−1^ at a power density of 478.5 W kg^−1^, and its cycling stability was 88.7% after 6000 cycle operations. The excellent performance of inorganic boron-based supercapacitors suggests that they should be ideal choices for the flexible anode materials of high-performance batteries and supercapacitors.

#### 3.1.3. Surface Electrical Transport Property

The surface states of topological insulators are always concealed by the dominant bulk conduction due to the existence of massive dopants or defects. In this situation, the topological Kondo insulator (TKI) becomes very attractive because the strongly correlated electron system can assure that the surface conduction dominates over their electrical transport. As typical TKI materials, rare-earth hexaborides (such as SmB_6_ and YbB_6_) have been the research goals for theoretical calculations and experiments [22,75,76]. Wirth et al. [77] observed that the surface states of SmB_6_ dominated the electron conduction when the temperature was below about 7 K due to a suppressed Kondo effect at the surface. Furthermore, Wirth et al. [78] elucidated the effect of magnetic or non-magnetic impurities on the topological surface states of SmB_6_ by the combination of local STS and macroscopic transport measurements, unveiling that the local shielding lengths of the surface states can change with the magnetic properties of the substitutes. Simultaneously, Yu, He, and Liu et al. [22,79] respectively investigated the local and nonlocal magnetotransport properties of individual SmB_6_ nanowires (NWs). Yu et al. observed an obvious transition from negative to positive magnetoresistance (MR) as the bias current increased in a single SmB_6_ nanowire, as shown in Figure 13. He and Liu et al. found that the hysteretic magnetoresistance effect will emerge in TKI SmB_6_ nanowires with diameters less than 58 nm. A non-monotonically temperature-dependent positive magnetoresistance is observed at intermediate temperatures because of the strong magnetism on the narrow nanowire’s surface [22], suggesting that impurity band conduction may exist in the SmB_6_ nanowire (Figure 14). Meanwhile, YbB_6_ is also predicted to be a TKI. Shi and Feng et al. [80,81] investigated the surface and bulk gap structures of YbB_6_ by angle-resolved photoemission spectroscopy (ARPES). Shi disclosed that the *f*-orbitals of YbB_6_ are fully occupied, and Yb exists in bivalence states, which is different from mixed-valence Sm in SmB_6_. Furthermore, the metallic surface states of YbB_6_ are believed to be topological surface states, which are spin-polarized in plane and locked to the crystal momentum. Feng et al. [81] directly observed the bands around the time-reversal invariant momenta exhibiting an obvious linearly dispersive relationship. The in-gap states possess the chirality of orbital angular momentum, which is attributed to the chiral spin texture. This unveils that YbB_6_ is a moderately correlated topological insulator.

In addition, 2D boron-based nanomaterials (borophene and boronphane) were predicted to be ideal Dirac materials, which exhibit clear linear energy dispersion characteristics at the Fermi level and have the Dirac cones in the band diagram based on the first-principle calculations [63,64,65]. For monolayer β_12_ borophene, Ezawa [82] proposed the existence of triplet fermions at the high-symmetry points, and there should be no loop encircling the triple-band bonding point without touching the Fermi surface. Furthermore, bilayer borophene is also a Dirac material, while few-layered borophene retains robust metallic characteristics owing to its multiple band interactions [83]. Compared to monolayer borophene with a high Fermi velocity close to graphene (8.2 × 10^5^ m/s) [84], borophane was calculated to possess a Fermi velocity (3.5 × 10^6^ m/s) that was two to four times larger than graphene, implying that they should be promising systems for high-speed electronic or optoelectronic devices. However, to date, all of the research studies on the surface topological behaviors of 2D boron-based nanomaterials have focused on the theoretical predictions, which are indeed short of the experimental evidence.

### 3.2. Optical Properties

#### 3.2.1. Optical Absorption

Rare-earth hexaborides usually have high absorbance in the visible and near-infrared (NIR) wavelength ranges [85,86,87], which can vary with their lattice constants (Figure 15a). Moreover, LaB_6_ nanoparticles with high conductivity create a promising metal-like plasmonic material that resembles Au or Ag nanoparticles. In this situation, LaB_6_@SiO_2_ (core/shell) nanoparticles were used as the photothermal catalysts in the reduction of 4-nitrophenol [87]. By the surface decoration of Au nanoparticles, LaB_6_@SiO_2_/Au composite nanoparticles (Figure 15b) exhibited a better photothermal conversion capacity, because the existence of Au nanoparticles offered more active spots on the catalyst surface and enhanced the temperature of the reaction process [88]. As observed in Figure 15, Sani et al. found that the absorbance coefficient of LaB_6_ (0.7) is comparable to the advanced solar absorber materials, such as SiC (0.8) and HfB_2_ (0.5) [89]. The absorption valleys of REB_6_ (CeB_6_, PrB_6_, and NdB_6_) nanostructures (Figure 15d) are respectively located at the visible regions of 670 nm, 785 nm, and 800 nm, suggesting that the rare-earth hexaborides may have potential applications as a sunlight absorber in the visible and NIR ranges [90].

#### 3.2.2. Photosensitive Properties

Boron and rare-earth boride nanostructures generally have small bandgaps and high refractive indices. Recently, our group observed the strong anisotropic light scattering behaviors and photocurrent response of tetragonal single crystalline boron nanowires (BNWs) in the visible region [48]. As indicated in Figure 16a,b, second harmonic generation (SHG) effects were discovered in a single boron nanowire under femtosecond laser irradiations. The individual BNW device has high device sensitivity (20), large responsivity (12.12 A W^−1^), and a fast on–off response (18 ms) (Figure 16c,d). Also, Yu et al. [91] found that a single SmB_6_ nanowire exhibited self-powered photodetector performances, in which the photovoltaic coming from the scanning photocurrent microscopy was responsible for the photocurrent. Moreover, the SmB_6_ nanowire device has a broadband response from 488 nm to 10.6 μm at room temperature, and an on/off ratio of about 100, as seen in Figure 17. Their responsibility and specific detectivity were respectively 1.99 mA W^−1^ and 2.5 × 10^7^ Jones. As observed in Table 4, the boron-based nanostructures have exhibited comparable photosensitive behaviors with many other nanostructures with excellent working performances, suggesting that they are promising candidates for future high-performance photodetectors.

## 4. Outlook and Conclusions

In conclusion, we have reviewed the recent progress of inorganic boron-based nanostructures (boron monoelement and rare-earth borides). As described above, CVD, laser ablation, magnetron sputtering, and thermal evaporation methods are the usual fabrication techniques of 1D boron-based nanostructures, whereas the MBE, CVD and liquid-phase exfoliation methods are more convenient for the synthesis of 2D boron-based nanostructures. The boron-based nanostructures have potential applications in field emission, supercapacitor, optical absorption, and photodetectors. Among them, 1D rare-earth boride nanostructures are promising as cold cathode electron sources, because they have a lower work function, higher melting point, metallic conductivity, and longer duration time, as well as a larger emission current. Since they belong to strongly correlated electron systems, SmB_6_, CeB_6_, and YbB_6_ TKIs are ideal platforms for investigating the surface quantum behaviors in condensed matter physics and material sciences. For 2D borophene or borophane, they have distinct advantages in flexible energy conversion devices or high-speed electronic devices, because they have metallic behaviors, a larger specific surface area, a higher Young’s modulus, and extremely high Fermi velocity. Also, 1D boron-based nanostructures exhibit strong light-absorption behaviors or rapid photocurrent responses, which suggests they are good candidates for flexible photodetectors at the visible and infrared ranges.

Although the research on the synthesis and optoelectronic properties of inorganic boron-based nanostructures has gained many achievements, there are still many essential and challenging issues that to be further explored. Designing and fabricating the LaB_6_ or CeB_6_ nanostructure-based cold cathode electron source remains a big challenge for the researchers. As a potential electron source, many important problems need to be solved in advance, such as a long-term and stable working performance at high current, long duration at severe conditions, a simple and cheap technique for a high-yield synthesis method, and low-power operation. For topological Kondo insulators, the fabrication of high-quality rare-earth boride nanostructure single crystals and exploration of the modulation techniques of their surface electrical transport behaviors are very important for their actual applications, which are still hard to solve. As for 2D boron-based nanostructures, the synthesis of a large-area single crystalline thin film with controllable layer thickness and chirality is a vital issue for their future applications. Another issue to explore and investigate is their optical and electrical properties in experiments, which are very essential because the current research studies mainly focus on the theoretical predictions. Although there are many challenges for the applications of inorganic boron-based nanostructures, they should have a very promising future, because they have exhibited enough fascinating properties in both experiments and theoretical calculations.

## Figures and Tables

**Figure 1 nanomaterials-09-00538-f001:**
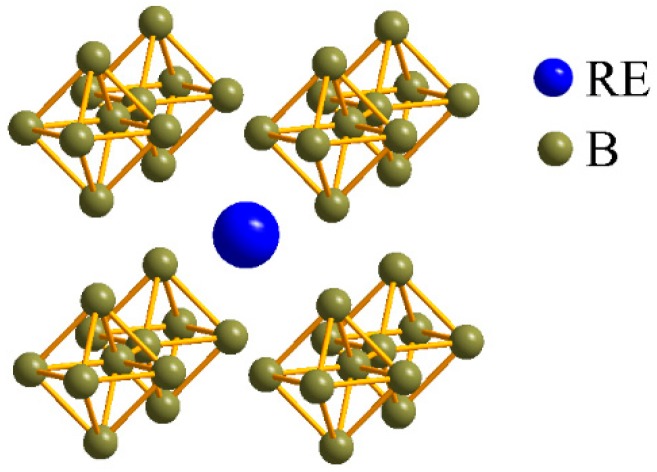
Typical lattice structure of rare-earth boride (REB_6_).

**Figure 2 nanomaterials-09-00538-f002:**
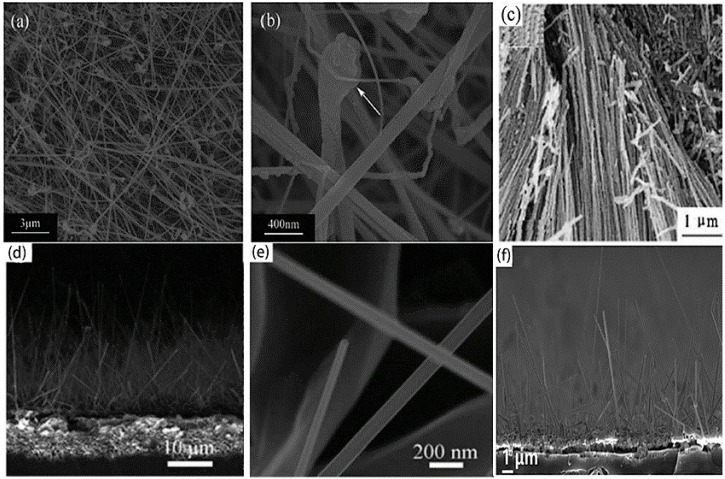
(**a**,**b**) Typical low-magnification and high-magnification SEM images of the PrB_6_ nanowires [39]. Copyright 2014, Elsevier. (**c**) Low-magnification SEM images of the NdB_6_ nanowires [41]. Copyright 2013, Elsevier. (**d**,**e**) Representative morphologies of the LaB_6_ nanowires [44]. Copyright 2017, Royal Society of Chemistry. (**f**) Side-view SEM image of the SmB_6_ nanowires [45]. Copyright 2017, IOP Publishing.

**Figure 3 nanomaterials-09-00538-f003:**
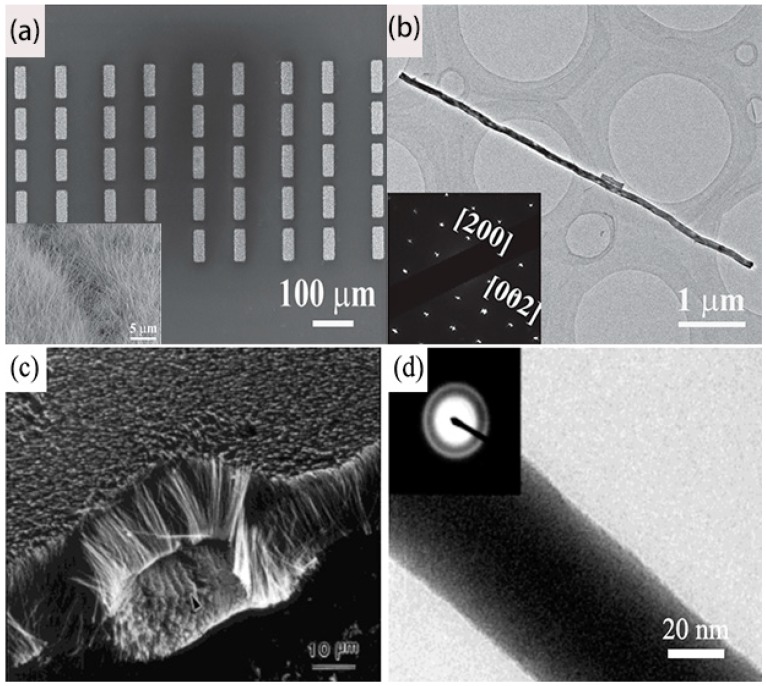
(**a**) Low-magnification SEM images of large area boron nanowire (BNW) arrays, and the inset gives their high-resolution image. (**b**) Typical HR-TEM image and SAED pattern of the single-crystalline BNW [51]. Copyright 2014, Wiley. (**c**,**d**) SEM and TEM image of the amorphous BNWs. Inset, SAED pattern taken from the nanowire showing some amorphous halo rings [46]. Copyright 2001, Wiley.

**Figure 4 nanomaterials-09-00538-f004:**
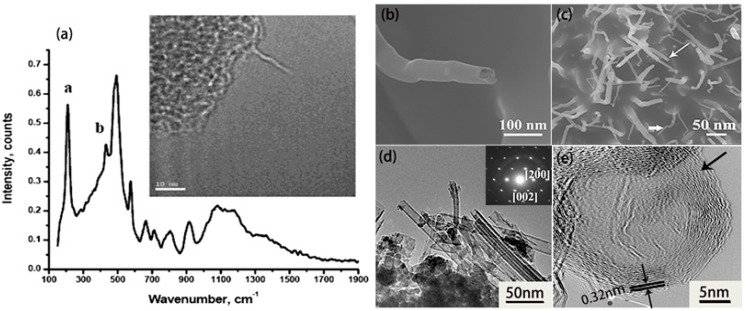
(**a**) Raman spectrum and TEM image (inset) of a single-wall boron nanotube [52]. Copyright 2004, American Chemical Society. (**b**,**c**) Low-resolution and high-resolution SEM images of the boron nanotubes (BNTs) (white arrows) at the initial growth stage. (**d**) TEM image of the BNTs. The inset is the corresponding SAED pattern. (**e**) The HRTEM image of the top of the BNT [53]. Copyright 2010, Royal Society of Chemistry.

**Figure 5 nanomaterials-09-00538-f005:**
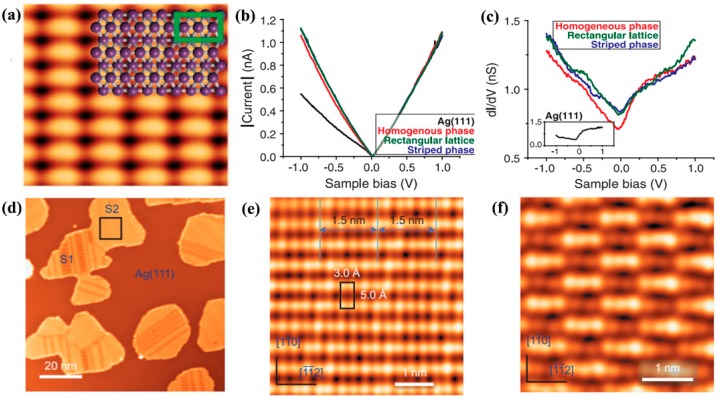
(**a**) Theoretical model and experimental STM images of borophene (V_sample_ = 0.1 V, I_tunnel_ = 1.0 nA). (**b**,**c**) I–V curves and dI/dV spectra of borophene [24]. Copyright 2015, Science. (**d**) STM image of boron sheets after 650 K annealing, in which two different phases are respectively labeled ‘S1’ and ‘S2’. (**e**) The S1 unit cell and the 1.5-nm stripes are respectively marked by a black rectangle and the solid lines. (**f**) High-resolution STM images of the S2 phase. Bias voltages of STM images: −4.0 V (**d**), 1.0 V (**e**,**f**) [25]. Copyright 2016, Springer Nature.

**Figure 6 nanomaterials-09-00538-f006:**
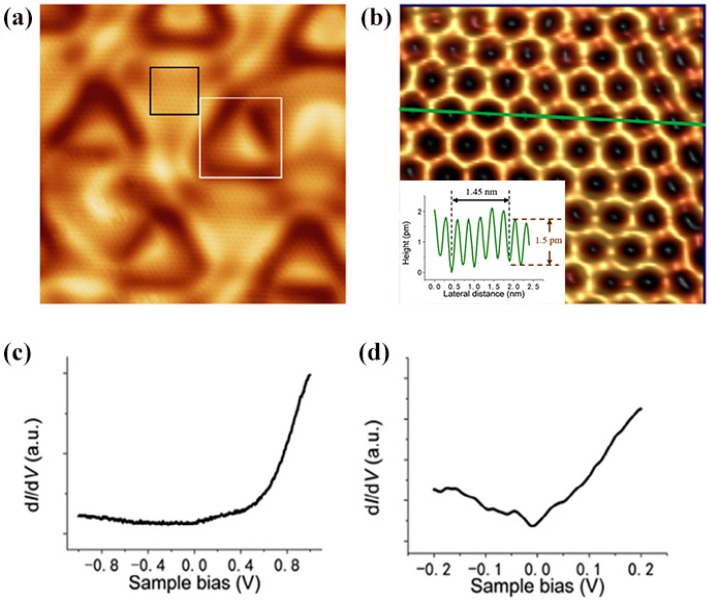
(**a**) STM image (15 nm × 15 nm) revealing the long-period configuration with triangular corrugation. (**b**) A high-resolution STM image (2.4 nm × 2.4 nm) of the area marked by the black rectangle in (**a**), showing a flat honeycomb lattice. Inset is a line profile along the green line. (**c**,**d**) The dI/dV curves taken on a borophene surface with a different bias voltage range. The scanning parameters for (**a**,**b**) are: sample bias −11 mV, I = 130 pA [59]. Copyright 2018, Elsevier.

**Figure 7 nanomaterials-09-00538-f007:**
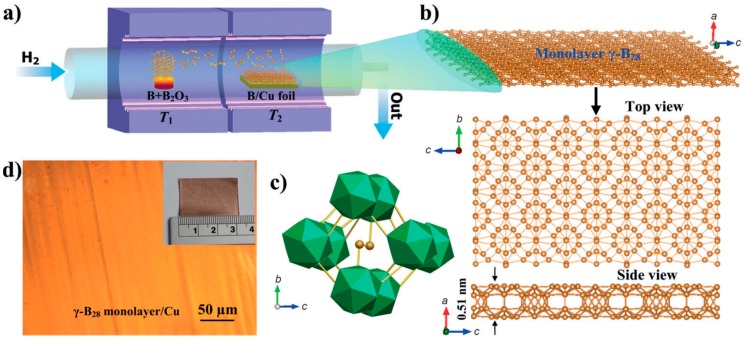
(**a**) Schematic diagram of the two-zone CVD furnace for two-dimensional (2D) γ-B_28_ film. (**b**) Top and side views of the borophene monolayer. (**c**) The corresponding polyhedral structure of the basic unit cell in bc projection, in which boron atoms (orange spheres) form the dumbbells. (**d**) Optical image of a monolayer on Cu foil [23]. Copyright 2015, Wiley.

**Figure 8 nanomaterials-09-00538-f008:**
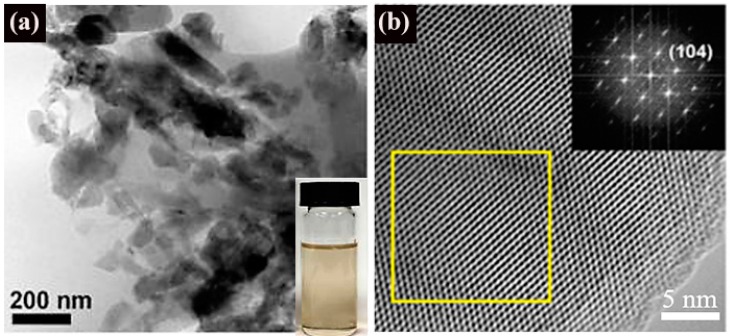
(**a**) Typical TEM images of the few-layer B sheets prepared by tip sonication. The inset is a photograph of B-sheet dispersions and sonication after 4 h. (**b**) Typical TEM image of the few-layer B sheets. The inset shows the corresponding FFT pattern [62]. Copyright 2018, American Chemical Society.

**Figure 9 nanomaterials-09-00538-f009:**
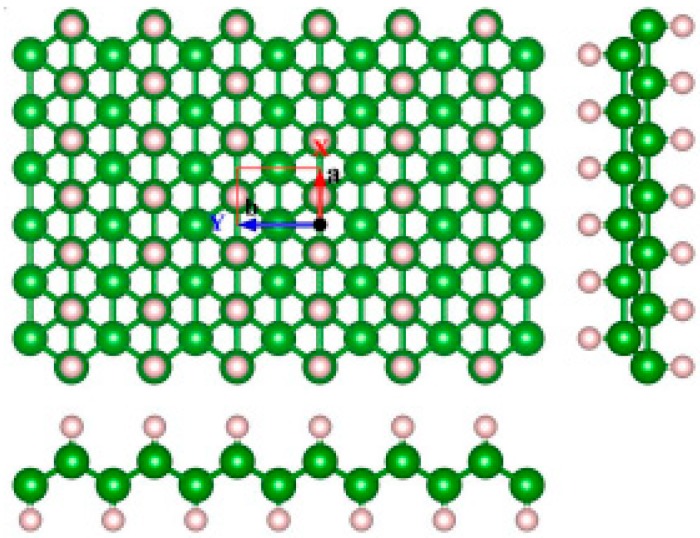
Top and side views of the optimized configuration of borophane. The unit cell is marked by a red box, in which the green and white balls respectively represent B and H atoms [64]. Copyright 2016, Royal Society of Chemistry.

**Figure 10 nanomaterials-09-00538-f010:**
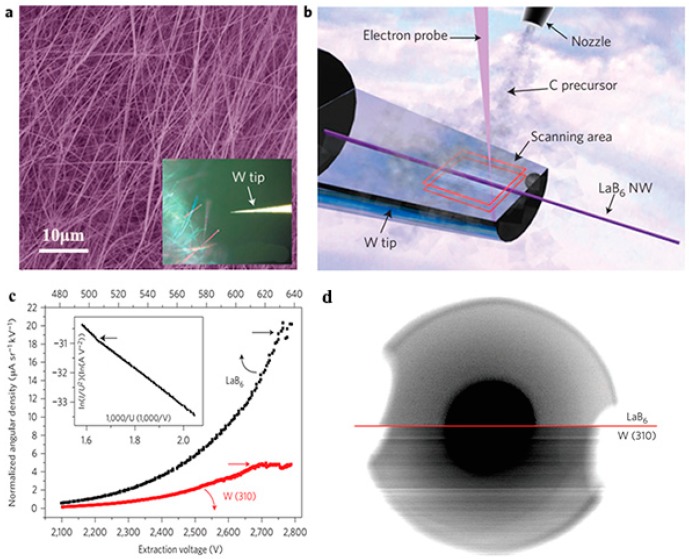
(**a**) SEM image of LaB_6_ nanowires (NWs) on a monocrystalline LaB_6_ (100) substrate. The inset shows the optical image in which a LaB_6_ nanowire is picked up by a W tip. (**b**) Schematic diagram showing the electron beam deposition process of fixing a LaB_6_ NW onto a W needle. (**c**) The dependence of normalized angular current density on extraction voltage with maximum value limited by emission instability. The inset gives their corresponding F–N plots, showing linearity with a transition point at high current density. (**d**) Signal noise comparison for imaging the acceleration electrode in the SEM objective lens using the LaB_6_ NW emitter and W emitter, respectively [68]. Copyright 2016, Springer Nature.

**Figure 11 nanomaterials-09-00538-f011:**
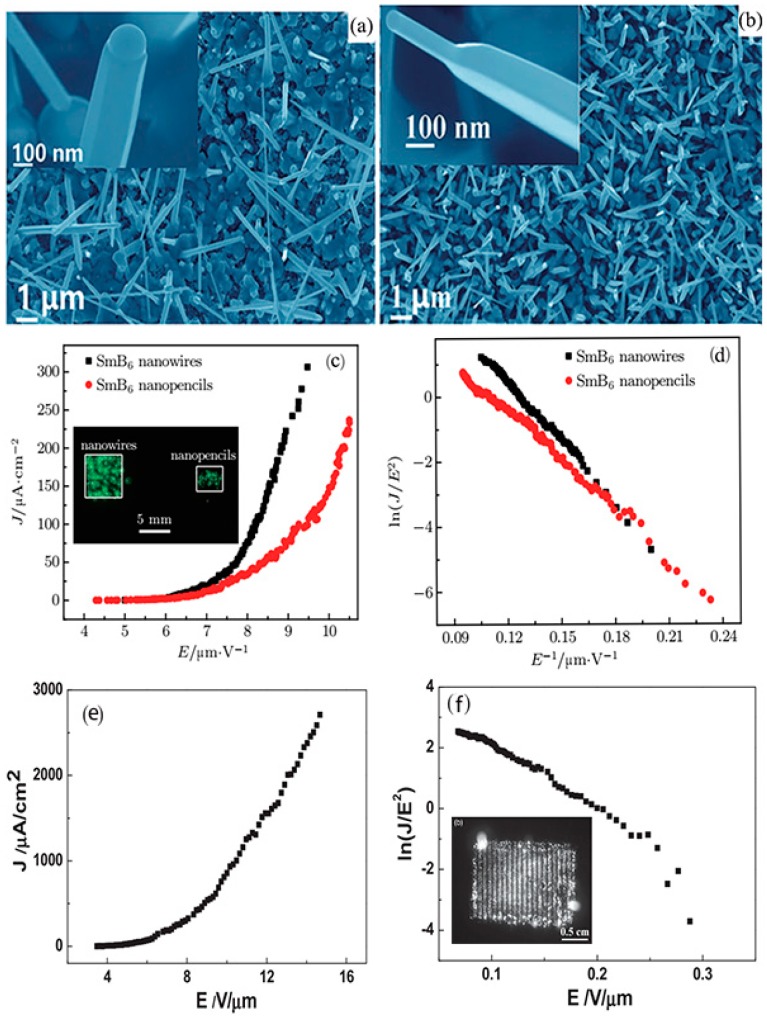
(**a**,**b**) Top-view SEM images of the SmB_6_ nanowires and nanopencils, respectively. The insets are their corresponding high-magnification cross-sectional images. (**c**,**d**) J–E curves and FN plots of the SmB_6_ nanowires and nanopencils. Their emission images are given in the inset [45]. Copyright 2017, IOP Publishing. (**e**,**f**) Typical J–E curves and FN plots of large-area boron nanowire patterns, and the inset is their corresponding emission image [51]. Copyright 2014, Wiley.

**Figure 12 nanomaterials-09-00538-f012:**
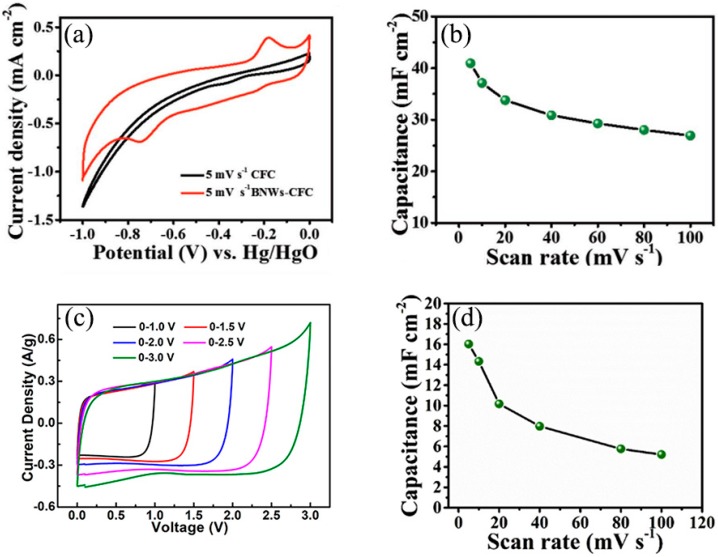
(**a**) Comparison of CV curves of carbon fiber cloth (CFC) and boron nanowire–carbon fiber cloth (BNWs–CFC) at a scan rate of 5 mV s^−1^. (**b**) Areal capacitances of BNWs–CFC at different scan rates [73]. Copyright 2018, Wiley. (**c**) CV curves collected under various voltage windows at a scan rate of 10 mV s^−1^ of B sheets [62]. Copyright 2018, American Chemical Society. (**d**) Plots of scan rates against the areal capacitances of LaB_6_–CFC electrodes [74]. Copyright 2018, Elsevier.

**Figure 13 nanomaterials-09-00538-f013:**
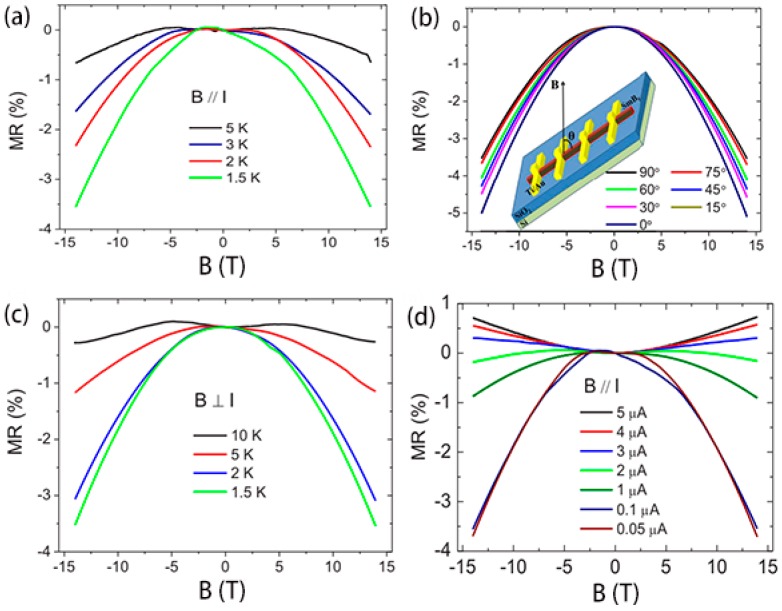
(**a**) The magnetoresistance (MR) for parallel magnetic field at different temperatures. (**b**) The MR measured at 1.5 K and under the magnetic field with different directions. The inset schematically shows the angle θ between the direction of the magnetic field and the longitudinal direction of the nanowire. (**c**) The MR for perpendicular magnetic fields at different temperatures. (**d**) The MR at 1.5 K for different bias currents ranging from 0.05 μA to 5 μA [79]. Copyright 2017, American Physical Society.

**Figure 14 nanomaterials-09-00538-f014:**
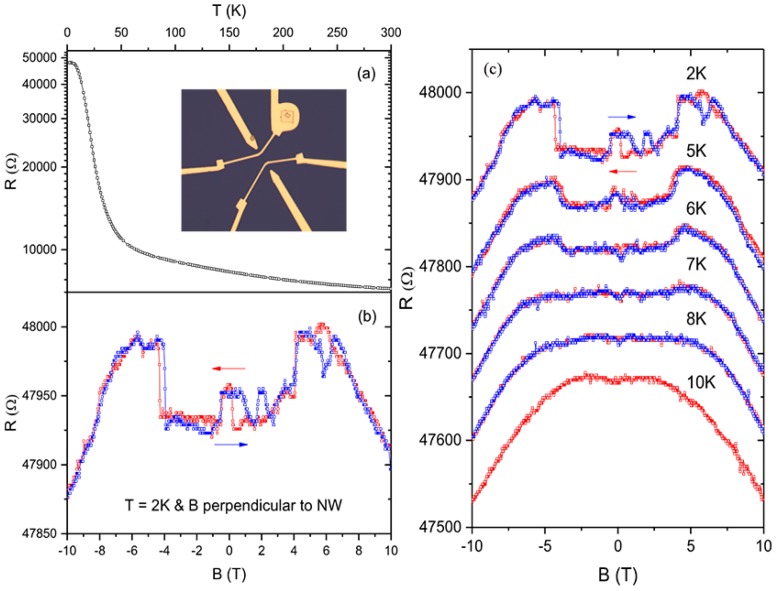
(**a**) The temperature dependence of resistance of a single SmB_6_ nanowire with a D value of 45 nm. The inset shows the fabricated four-probe device. (**b**) Double-sweep magnetoresistance curve with B perpendicular to the NW and T = 2 K. The field sweeping directions are indicated by arrows. (**c**) Double-sweep MR curves of a SmB_6_ NW in perpendicular magnetic fields at different temperatures. The NW diameter is 45 nm, and the field-sweeping directions are indicated by arrows. Curves are offset vertically for comparison. At T = 10 K, the MR curve was only measured in the sweep-down direction [22]. Copyright 2018, Wiley.

**Figure 15 nanomaterials-09-00538-f015:**
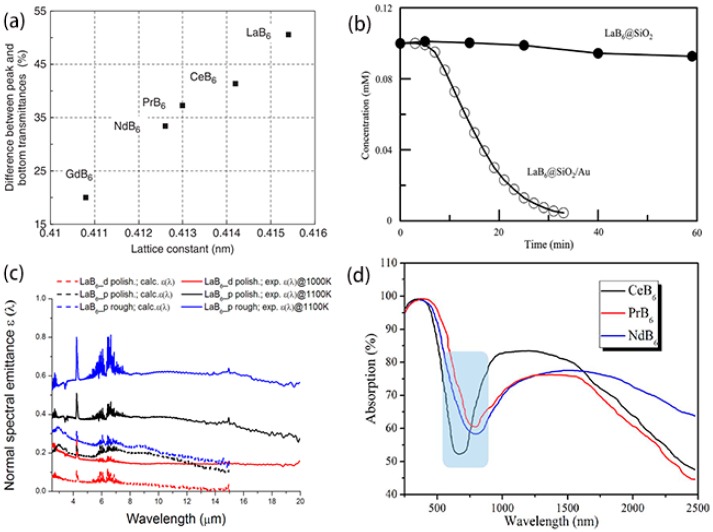
(**a**) Near-infrared absorption of rare-earth hexaboride nanoparticles with regard to their lattice constants [85]. Copyright 2008, Wiley. (**b**) Variation of 4-nitrophenol concentration with time by LaB_6_@SiO_2_ and LaB_6_@SiO_2_/Au composite nanoparticles [88]. Copyright 2013, Elsevier. (**c**) Calculated (dashed lines) and experimental emittance (solid lines) of LaB_6_ samples [89]. Copyright 2017, Springer Nature. (**d**) The optical absorption of REB_6_ nanostructures at visible and near-infrared ranges [90]. Copyright 2017, Elsevier.

**Figure 16 nanomaterials-09-00538-f016:**
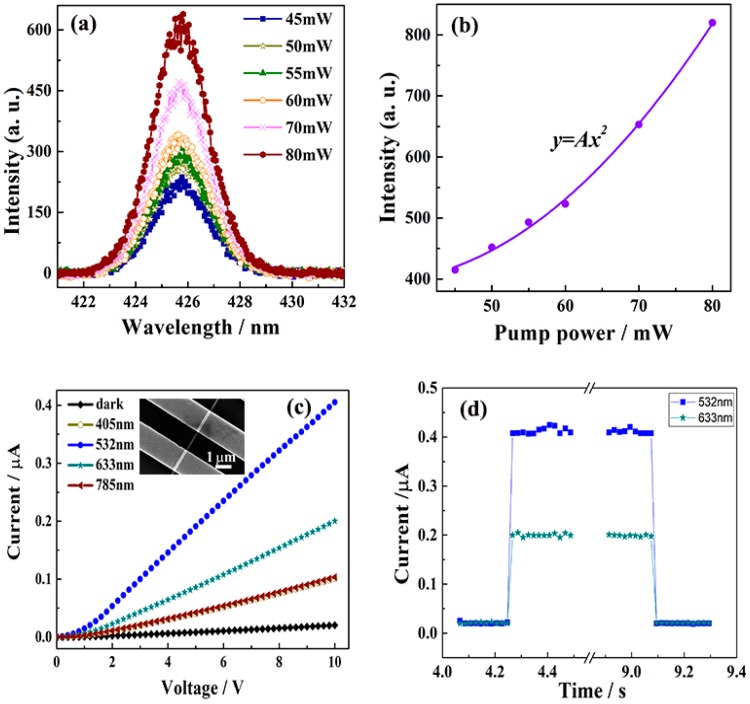
(**a**) Second-harmonic generation spectra of the BNW film under 850-nm illumination with different pump powers. (**b**) The curves of the second harmonic generation (SHG) peaks’ intensity versus the pump powers. (**c**) The photocurrent-voltage curves of an individual BNW under different irradiations. The inset is the nanodevice structure. (**d**) The fast on–off curves of the BNW under 532-nm and 633-nm irradiations, respectively [50]. Copyright 2018, Wiley.

**Figure 17 nanomaterials-09-00538-f017:**
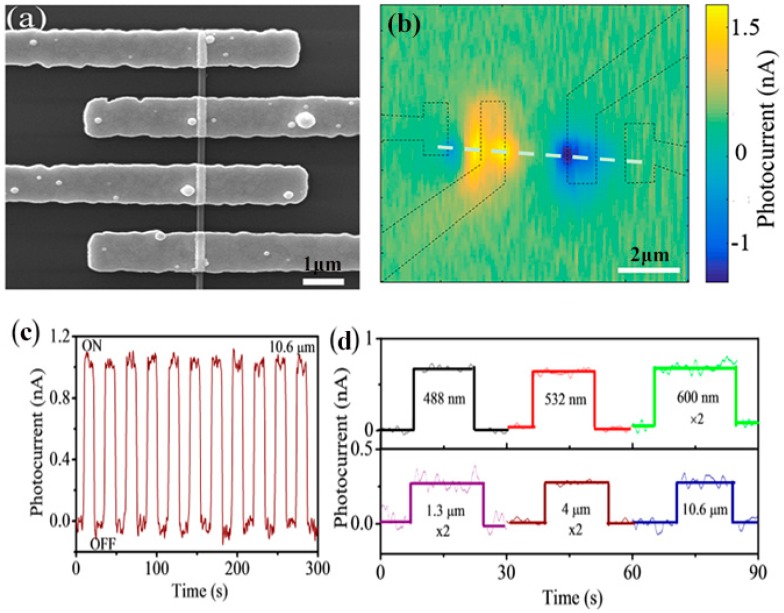
(**a**,**b**) SEM image and scanning photocurrent mapping of a single SmB_6_ nanowire photodetector. (**c**) The time-dependent photocurrent measurement on the nanowire photodetector under 10.6-μm irradiation. (**d**) Room temperature photoresponse of the SmB_6_ device under photoexcitation with different excitation wavelengths (as marked) [91]. Copyright 2018, American Institute of Physics.

**Table 1 nanomaterials-09-00538-t001:** The summary of the synthesis methods of different rare-earth boride nanostructures. CVD: chemical vapor deposition.

Materials	Source Materials	Method	T/℃	Catalyst	Ref.
LaB_6_ nanowire	LaCl_3_·7H_2_O, B_2_H_6_	CVD	930	free	[35]
LaB_6_ nanoneedle	LaCl_3_·7H_2_O, B_2_H_6_	CVD	970	free	[35]
PrB_6_ nanowire	Pr powders, BCl_3_ gas	CVD	1000-1150	free	[38]
NdB_6_ nanowire	Nd powders, BCl_3_ gas	CVD	1150	free	[40]
LaB_6_ nanowire	B, B_2_O_3_, LaCl_3_ powders	CVD	1100	Ni-assisted	[44]
SmB_6_ nanowire	B, B_2_O_3_ powders, Sm film	CVD	1100	Ni-assisted	[45]
SmB_6_ nanowire	Sm, H_3_BO_3_, Mg and I_2_ powders	Hydrothermal reaction	220-240	I_2_-assisted	[39]
LaB_6_ nanowire	H_3_PO_4_, C_2_H_5_OH,LaB_6_ target	Electrochemical etching	2	free	[43]

**Table 2 nanomaterials-09-00538-t002:** The summary table of the synthesis methods of borophene. MBE: molecular beam epitaxy.

Method	Temperature	Substrate	Structural Configuration	Source Materials	Sample Area
CVD [23]	1100 ℃	Cu foil	γ-B_28_	B, B_2_O_3_ powders	Nanometer
MBE [24,25,58]	300–750 ℃	Ag (111)	β_12_ and χ_3_	B powder	Nanometer
MBE [59]	230 ℃	Al (111)	honeycomb	B powder	Nanometer
MBE [60]	300/490 ℃	Cu(111)	β_12_ and χ_3_	B powder	Micrometer
Plasma-assisted ion implantation [61]	800 ℃	Si (001)	β	B powder	Nanometer
Liquid-phase exfoliation [62]	N/A	N/A	β-rhombohedral	B powder	Nanometer

**Table 3 nanomaterials-09-00538-t003:** Comparison table of the FE properties of some nanomaterials with excellent emission performances.

Nanomaterials	Turn-on Field Vμm^−1^	Threshold Field Vμm^−1^	Emission Current Fluctuation	Ref.
Mo nanoscrew	1.65	2.4	0.46%, 1 h, 50 mA/cm^2^	[69]
Carbon nanotube	3.2	5.8	25%, 20 h, 260 mA/cm^2^	[70]
ZnO nanobelt	6.6	8.5	14%, 16 h, 7.4 mA/cm^2^	[71]
SiC nanowire	0.9	1.7	3%, 24 h, 5 mA/cm^2^	[72]
LaB_6_ nanowire	1.9	2.9	1.2%, 0.5 h, 2.6 3mA/cm^2^	[44]
Boron nanowire	4.3	10.4	5.6%, 8 h, 1 mA/cm^2^	[51]

**Table 4 nanomaterials-09-00538-t004:** The comparable table of the working performances of various nanodevices. UV: ultraviolet, MIR: middle infrared.

Nanostructures	Operation Voltage [V]	Device Sensitivity (I_P_/I_D_)	Photoresponsivity (R_λ_) [A W^−1^]	Detection Range	Response Time	Ref.
ZnO nanowire	2	8	N/A	UV	50 s	[92]
AlN nanowire	40	20	2.7×10^6^	UV-Visible	10 ms	[93]
GaN nanowire	0	13	N/A	UV	0.53 s	[94]
B nanowire	10	20	12.12	Visible	18 ms	[48]
SmB_6_ nanowire	0	100	1.99×10^-3^	Visible-MIR	N/A	[91]

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
