# Peer review of "Inorganic Boron-Based Nanostructures: Synthesis, Optoelectronic Properties, and Prospective Applications"

_nanomaterials, 2019, doi:10.3390/nano9040538_

Reviewer 1 Report

The authors did a nice review of the recent progress in the synthesis methods and optoelectronics properties of one-dimensional (1D) rare-earth boride nanowire, boron monoelement nanowire and nanotube as well as two-dimensional (2D) borophene and borophane. The subject is new and interesting to review. It is worth to publish this review after minor revisions.

The authors need to double check and correct the grammar of the text. For example, the authors sometimes use the past sometimes the present.
Double check that all references are cited within the text, and that all citations within the text have a corresponding reference. Double check the spelling of the author names, the format of the references.

Please double check the manuscript for abbreviations. Abbreviations must be spelled out the first time they are mentioned in the abstract and starting again with the introduction section:
like “FE”

In table 2, Temperature should be together

Line 237: “In experiments?” which experiments the authors mean?

Line 279, “Soon afterwards Teo” should be “Soon afterwards, Teo”

No reference is cited in “Feng et al. directly” and the sentence need to be corrected.

Line 337, the authors should rewrite the sentence “It’s believed that the LaB6 nanoparticles may have the largest NIR absorption capacity, which should”.

Line 346, “material” should be “materials”

Line 369: “was” should be “were”

The units are sometimes with space sometimes without like for “Vμm-1

Figure 6: the description for (c) is missing

Author Response

Replies to the first reviewer comments are as follows:

1.     Reply to Query 1(The authors need to double check and correct the grammar of the text. For example, the authors sometimes use the past sometimes the present.)

   We are thankful for the beneficial suggestion of the reviewer on our work. We have carefully smoothed our manuscript based on the referee’s suggestion in our revised paper.

 2.     Reply to Query 2 (Double check that all references are cited within the text, and that all citations within the text have a corresponding reference. Double check the spelling of the author names, the format of the references.)

 We are very appreciated for the helpful advice of the reviewer on our paper. We have double checked the spelling of the author names and the format of the references to assure they are all correct according to the referee’s suggestion.

 3.      Reply to Query 3(Please double check the manuscript for abbreviations. Abbreviations must be spelled out the first time they are mentioned in the abstract and starting again with the introduction section: like “FE”)

We thanks a lot for the helpful suggestions of the referee on our article. Based on the suggestion, we have supplied the full name of the abbreviations at the first time they are mentioned in the paper in our revised version. The revisions can be found in the 11th , 14th and 15th lines of Page 1 in our revised paper.

 4.      Reply to Query 4(In table 2, Temperature should be together)

We are thankful for the beneficial suggestion of the reviewer on our work. We have combined the MBE growth temperatures on Ag (111) of into together in Table 2, as seen on Page 9 in the revised version.

  Table 2. The summary table of the synthesis methods of borophene.

Method

Temperature

Substrate

Structural   Configuration

Source

materials

Sample Area

CVD [23]

1100℃

Cu foil

γ-B28

B, B2O3   powders

Nanometer

MBE   [24-25],[58]

300-750℃

Ag (111)

β12 and χ3

B powder

Nanometer

MBE [59]

230℃

Al (111)

honeycomb

B powder

Nanometer

MBE [60]

300/490℃

Cu(111)

β12 and χ3

B powder

Micrometer

Plasma-assisted   [61]

800℃

Si (001)

β

B powder

Nanometer

Liquid-phase    exfoliation [62]

N/A

N/A

β-rhombohedral

B powder

Nanometer

 5.      Reply to Query 5(“In experiments?” which experiments the authors mean?)

    We are very thankful for the careful inspection of the reviewer on our paper. We have corrected “In experiment” into “In FE experiments in the modified SEM system [68]” according to the suggestion, which can be seen in the 265th line of Page 10 in our revised paper.

6.      Reply to Query 6(“Soon afterwards Teo” should be “Soon afterwards, Teo”. No reference is cited in “Feng et al. directly” and the sentence need to be corrected.)

We are very appreciated for the helpful advice of the reviewer on our paper. We have added the comma “,” into this sentence based on the referee’s suggestion, as shown in the 309th line of page 12 in our revised paper.

In addition, the reference of “Feng et al. [81] directly…” has been added into the 346th line of page 14 in our revised paper according to the suggestion.

 7.      Reply to Query 7(the authors should rewrite the sentence “It’s believed that the LaB6 nanoparticles may have the largest NIR absorption capacity, which should”.)

We are very grateful for the helpful advice of the reviewer on our manuscript. We have correct it into “in which the LaB6 nanoparticles may have the largest NIR absorption capacity.”, as seen in the 385th line of Page 15 our revised paper.

8.       Reply to Query 8(“material” should be “materials”, “was” should be “were”, The units are sometimes with space sometimes without like for “Vμm-1)

We are very appreciated for the helpful advice of the reviewer on our paper. We have carefully corrected these mistakes in our revised paper according to the suggestions, which can be respectively seen in line 391 of Page 15, line 415 of Page 16 and lines 257-259 of Page 10.

9.      Reply to Query 9 (Figure 6: the description for (c) is missing)

We are very thankful for the beneficial suggestion of the reviewer on our work. Based on the suggestion, we have added the caption of Figure (c) “(c, d) The dI/dV curves taken on borophene surface with different bias voltage range.into the 21102th line on Page 8 in our revised paper.

Reviewer 2 Report

This is a good topic to review, but given the wide diversity of methods, materials, characterization techniques, and applications, it is difficult to cover with coherence and  insight.  In this regard, the authors have not done a particularly good job.

The authors' style is to essentially list one experiment after another.  They briefly describe how the experiment was performed and the outcome of the experiment.  It is like reading a list instead of a narrative.

For the synthesis experiments, it's not clear what they main objectives are other than to produce a specific composition.  What are the optimum structures researchers are trying to achieve?  Presumably the researchers want to control whether the nanostructures are amorphous or crystals, have specific crystal structures, have deviations from stoichiometry, the aspect ratios for nanowires and nanotubes, and thickness for layers.  Provide context: were the researchers successful?  It's unclear whether the technology exist to achieve specific goals.

The paper starts poorly with the false statement "Boron is very special because it is the only electron-deficient element and the lightest solid element in the world."  It is not the only electron-deficient element: this is true of all of the halides (F, Cl, Br,I).  It is not the lightest solid element either: lithium has lower density and is thus lighter.  

Continuing in the first paragraph, there is the statement "With the rapid development of information technology, large-scale and extremely large-scale integrated circuits consisted of high density nanodevices are highly desired for microelectronics industries."  While true, it has nothing to do with boron-base nanostructures, which are not being studied for microelectronics.

There seems to be little filtering or curating of information to bring a sense of purpose to this review.  Figure 2 is typical. Six images are presented of nanowires with three different compositions.  The description of this figure is limited to one sentence in the text body.  What is the point of this figure?  Is it that the morphologies and sizes are similar?  What about the orientation of these nanowires?  Are they all the same?

Terms are not well defined.  What meant by nanowire?  Is it crystalline or can it be amorphous?  What is a "nanopencil?"  

The review also suffers from its poor English.  The authors repeatedly use the phrase "as observed" over and over again. Words used repeatedly include "observed," "obtained," "shown," and "found."  This repetition make reading this paper very dull.  Suggestions to improve the English are included in the marked-up manuscript accompanying this review.

Reference numbers should immediately follow the phrase et al.   It should be "Xu et al [36]" and not the reference number at the end of the sentence. 

I urge the authors to seek to make this a more coherent paper.

Author Response

Replies to the second reviewer comments are as follows:

 1.     Reply to Query 1(For the synthesis experiments, it's not clear what they main objectives are other than to produce a specific composition. What are the optimum structures researchers are trying to achieve?  Presumably the researchers want to control whether the nanostructures are amorphous or crystals, have specific crystal structures, have deviations from stoichiometry, the aspect ratios for nanowires and nanotubes, and thickness for layers. Provide context: were the researchers successful?  It's unclear whether the technology exist to achieve specific goals.)

    We are very thankful for the beneficial advice of the reviewer on our paper. Based on the suggestion, we reorganize and correct the experimental descriptions of boron-based nanomaterials in the revision version, as seen in the following. The four descriptions “For actual cold cathode applications, the REB6 nanomaterials with higher crystallinity, lower work function and larger aspect ratio are highly demanded, which become the crucial goal for the researchers. As illustrated in Table 1, the synthesis method of REB6 nanostructures is generally chemical vapor deposition (CVD), hydrothermal reaction and electrochemical etching method.…In comparison with other two ways, CVD method is more convenient for the growth of REB6 nanomaterials because their morphology, crystal structure and stoichiometry are much easier to control by modulating the experimental parameters as well as high-quality cathode nanomaterial arrays can be successfully obtained.”, “At the viewpoint of future cold cathode applications of 1D boron nanostructures, the CVD method usually take advantages over other synthesis methods owing to its better morphology, aspect ratio, and crystallinity controllability. In addition, high-density 1D boron-based nanostructure arrays can be much easier to obtain on rigid or flexible substrates by CVD way, which is very beneficial for their anode applications in cells or photosensitive devices. ” and “To date, there are three synthesis methods of 2D boron-based nanostructures, which are respectively molecular beam epitaxy (MBE), liquid-phase exfoliation and CVD methods. The researchers devote to modulate the morphology, layer number and structure configuration of 2D boron-based nanostructures because of which directly care about their physical properties. ” and “As mentioned above, the MBE way is more suitable for fabrication of monolayer or few-layer borophene with very nice crystallinity while CVD way have advantages on the layer or surface configuration control of large-area borophene with different standing direction to substrate. High-yield borophene are suggested to adopt the sonication-assisted liquid-phase exfoliation method, which can be widely used in the industry application. It can be noted that only a few configurations of borophene are successfully prepared until now, which is far less than tens of configuration number in theoretical prediction. As a result, it still remain a big challenge for the researchers.” have been added into the 83th line of page 3, 153th line of page 6, 160th line of page 6 and 239th line of page 9 in the revised paper. 

 2.     Reply to Query 2(The paper starts poorly with the false statement "Boron is very special because it is the only electron-deficient element and the lightest solid element in the world."  It is not the only electron-deficient element: this is true of all of the halides (F, Cl, Br,I).  It is not the lightest solid element either: lithium has lower density and is thus lighter.)

 We are very appreciated for the helpful advice of the reviewer on our paper. According to the suggestion, we have revised this sentence into “Boron is very special because it is the only nonmetallic element in group III and the lightest nonmetallic element in the periodic table as well as has some excellent properties resemble to carbon. Moreover, boron possesses unique physical and chemical properties due to the B12 icosahedra structural unit by a particular 3-center 2-electron bond”, which can be seen in the 30th line on page 1 in our revised paper.

 3.     Reply to Query 3(Continuing in the first paragraph, there is the statement "With the rapid development of information technology, large-scale and extremely large-scale integrated circuits consisted of high density nanodevices are highly desired for microelectronics industries."  While true, it has nothing to do with boron-base nanostructures, which are not being studied for microelectronics.)

We thank a lot for the helpful suggestions of the reviewer on our article. Based on the suggestion, we have deleted these unsuitable descriptions in our revised manuscript.

 4.     Reply to Query 4(There seems to be little filtering or curating of information to bring a sense of purpose to this review.  Figure 2 is typical. Six images are presented of nanowires with three different compositions.  The description of this figure is limited to one sentence in the text body.  What is the point of this figure?  Is it that the morphologies and sizes are similar?  What about the orientation of these nanowires?  Are they all the same?)

 We are very grateful to the reviewer for the helpful suggestions on our article. We have added these detailed descriptions into our revised paper according to the suggestion, as seen as follows. The description “As observed in Table 1, the synthesis methods of REB6 nanostructures can be classified into three types, which are respectively chemical vapor deposition (CVD), hydrothermal reaction and electrochemical etching methods. Chi and Fan et al. [38-43] respectively applied a no-catalyst CVD method to synthesize different REB6 (PrB6, NdB6, CeB6, GdB6 and EuB6) nanowires at 1000–1150℃ by choosing rare-earth metal powders and B-contained gas (BCl3 or B2H6) as source materials, whose morphological characteristics can be found in Fig. 2. As seen in Figures 2a-c, single crystalline PrB6 and NdB6 nanowires have diameters of several tens of nanometers and length extending to several micrometers, which can have uniform diameter along the growth axis. Using CVD method, Xu et al. [35] reported the successful growth of the LaB6 1D nanoneedles with a diameter decreasing from the bottom to the top and nanowires with uniform diameter by changing the distances between precursors and substrate. By hydrothermal reaction way, Han et al. [39] reported a low-temperature CVD way to obtain single crystalline SmB6 nanowires growing along [001] direction, whose averaged diameter and length are respectively 80 nm and 4 μm.” has been added into the 85th line on page 3 in the revision version.

 5.     Reply to Query 5(Terms are not well defined.  What meant by nanowire?  Is it crystalline or can it be amorphous?  What is a "nanopencil?")

    We are very thankful for the beneficial suggestions of the referee on our paper. According to the suggestions, we have added the detail descriptions of the morphological characteristics of “nanowire”, “nanopencil” and “nanoneedles” in our revised paper, which can be seen as follows. The descriptions “…nanowires…, which can have uniform diameter along the growth axis…. 1D nanoneedles with a diameter decreasing from the bottom to the top” and “… the SmB6 nanopencils having a thick bottom and a sharp tip…” have been added into the 91th line of Page 3 and 269th line of Page 10 in the revision version, respectively.

 6.     Reply to Query 6(The review also suffers from its poor English.  The authors repeatedly use the phrase "as observed" over and over again. Words used repeatedly include "observed," "obtained," "shown," and "found."  This repetition make reading this paper very dull.  Suggestions to improve the English are included in the marked-up manuscript accompanying this review.)

We are very appreciated for the helpful advice of the reviewer on our paper. We have polished the paper very carefully to meet the publication requirements in our revised paper based on the suggestion.

 7.     Reply to Query 7(Reference numbers should immediately follow the phrase et al.   It should be "Xu et al [36]" and not the reference number at the end of the sentence.)

 We are thankful for the beneficial suggestion of the reviewer on our work. According to the suggestions, we have moved all the reference numbers of the text to just follow the researchers’ name in our revised paper. For example, “Chi and Fan et al.” [38-43] (the 87th line of Page 3) and “Xu et al. [35]” (the 93th line of Page 3), which can be found in the revised version.

Reviewer 3 Report

The present review focus on the synthesis and optoelectronic properties of Inorganic boron-based nanostructures. Boron-based nanostructures are very suitable for nanochemistry, electronic and thermal transport trandue to their low work-function. These properties have been studied for long time. However, the recent preparation of new 2D forms of inorganic boron species has renewed the interest on this material. In the last few years, many work has been done on this material but also many open questions still remain unresolved. For this reason, I consider that the present review is timely and very adequate to conduct further research on this element and related compounds, facing, in particular, a very important issue related to nanoforms of B, as it is stability.

Also, I have found the present review very well written and organized, well founded and connected to existing literature, able to call the attention of potential readers to this material and to serve as a platform for further studies. From all these reasons, I am glad to recommend the present manuscript for publication.

I only have a minor aspect that is addressed to extend the perspectives of the present review, which I would recommend to be introduced in a revised version of the manuscript: It has been mentioned in the manuscript that topological properties of Borophene is quite relevant, for a light material it is a surprising property. I would recommend to face the review of the electronic properties of this material, from the point of view of their topological perspectives, since they are important for new optoelectronic devices that may be designed with these materials.

Let me suggest you some references on this issue:

- X.-F.  Zhou,  X.  Dong,  A.  R.  Oganov,  Q.  Zhu,  Y.  Tian  and  H.-T.  Wang, Phys.  Rev.  Lett.,  2014, 112, 085502.
- X.-F. Zhou and H.-T. Wang, Adv. Phys.: X, 2016, 1, 412-424.
- H. Zhang, Y. Xie, Z. Zhang, C. Zhong, Y. Li, Z. Chen and Y. Chen, J. Phys. Chem. Lett., 2017, 8, 1707-1713.
- W. C. Yi, W. Liu, J. Botana, L. Zhao, Z. Liu, J. Y. Liu and M. S. Miao, J. Phys. Chem. Lett., 2017, 8, 2647-2653.
- M. Ezawa, Phys. Rev. B, 2017, 96, 035425.
- L. C. Xu, A. Du and L. Kou, Phys. Chem. Chem. Phys., 2016, 18, 27284-27289.
- Y. Jiao, F. Ma, J. Bell, A. Bilic and A. Du, Angew. Chem. Int. Ed. Engl., 2016, 55, 10292-10295.
- M. Nakhaee, S. A. Ketabi and F. M. Peeters, Phys. Rev. B, 2018, 98, 115413.
- M. Nakhaee, S. A. Ketabi and F. M. Peeters, Phys. Rev. B, 2018, 97, 125424.

Author Response

Replies to the third reviewer comments are as follows:

 1.     Reply to Query 1(I only have a minor aspect that is addressed to extend the perspectives of the present review, which I would recommend to be introduced in a revised version of the manuscript: It has been mentioned in the manuscript that topological properties of Borophene is quite relevant, for a light material it is a surprising property. I would recommend to face the review of the electronic properties of this material, from the point of view of their topological perspectives, since they are important for new optoelectronic devices that may be designed with these materials.)

 We are very thankful for the beneficial suggestion of the referee on our manuscript. As referred by the reviewer, boron-based nanomaterials should have fascinated and intriguing topological characteristics. Based on the suggestion, we have added some detailed descriptions on this issue, as seen as the following. The description “ In addition, 2D boron-based nanomaterials (borophene and boronphane) were theoretically predicted to be ideal Dirac materials, which exhibit clear linear energy dispersion characteristics at Fermi level and have the Dirac cones in the band diagram based on the first-principle calculations [63-65]. Intriguingly, Ezawa [82] discovered the triplet fermions in monolayer borophene, which is quite different from the Dirac fermions. Furthermore, bilayer borophene is also a Dirac material while few layers borophene retain robust metallic characteristic properties owing to multiple bands interaction [83]. Compared to monolayer borophene with high Fermi velocity close to graphene (8.2×105 m/s) [84], borophane was calculated to possess 2-4 times larger Fermi velocity (3.5×106 m/s) than graphene, which reveal they should be promising systems for high-speed electronic or optoelectronic devices. Besides, Majorana Fermions may exist at interface between pristine borophene and graphene or transition-metal dichalocogenides (TMDCs) [82], in which many novel physical properties or intrinsic mechanism can be explored. But to date, all of the researches on the surface topological behaviors of 2D boron-based nanomaterials are still focus the theoretical prediction, which are indeed short of the experimental evidence.” can be found in the 357th line of page 14 in our revised manuscript based on the important references. Moreover, these important references have been cited in our revised paper, which are respectively regarded as Refs. [64], [65], [82], [83] and [84] in References
